# Subjective Overview of Accelerated Aging in Schizophrenia

**DOI:** 10.3390/ijerph20010737

**Published:** 2022-12-31

**Authors:** Mary V. Seeman

**Affiliations:** Department of Psychiatry, University of Toronto, 260 Heath St. West, Suite #605, Toronto, ON M5P 3L6, Canada; mary.seeman@utoronto.ca

**Keywords:** schizophrenia, neurodevelopment, neurodegeneration, aging, antipsychotics

## Abstract

Schizophrenia, like many other human diseases, particularly neuropsychiatric diseases, shows evidence of accelerated brain aging. The molecular nature of the process of aging is unknown but several potential indicators have been used in research. The concept of accelerated aging in schizophrenia took hold in 2008 and its timing, pace, determinants and deterrents have been increasingly examined since. The present overview of the field is brief and selective, based on diverse studies, expert opinions and successive reviews. Current thinking is that the timing of age acceleration in schizophrenia can occur at different time periods of the lifespan in different individuals, and that antipsychotics may be preventive. The majority opinion is that the cognitive decline and premature death often seen in schizophrenia are, in principle, preventable.

## 1. Introduction

Accelerated or premature aging of body tissues can occur as a result of factors such as sun exposure, inflammation, smoking, diet, sleep habits, intake of alcohol and caffeine, stress, air pollution, dehydration, and sedentary lifestyle. It can also result from the expression of specific aging genes [1]. The phenomenon of accelerated aging has been observed in several neuropsychiatric disorders [2] and has been investigated from many perspectives. Here, are some examples: Breakdown of the circadian clock [3], Shortened telomere length [4,5], Multimorbidity [6], Brain structure and cognitive function [7] Oxidative stress and failure of protein repair [8], Metabolic factors [9], DNA methylation [10]. DNA epigenetic markers are modern tools that are beginning to be used more and more often as proxies for aging [11] but it is not clear whether this indicator, or any of the others, are accelerators of aging or responses to aging.

Age acceleration can occur in childhood, as in progeria [12], in adolescence, as has been reported in schizophrenia [13], or in late life as in Parkinson’s disease and Alzheimer’s dementia [14,15]. 

I have been interested for many years in the many parallels between how one copes with the symptoms of aging and how persons with schizophrenia cope with the symptoms of psychosis [16]. This earlier interest led me to ask the following questions of the increasingly large literature on the possibility of a process of age acceleration occurring in schizophrenia. This brief, selective review asks the following questions:Is schizophrenia characterized by accelerated aging?If so, when in the lifespan does the acceleration occur?Does schizophrenia drug treatment block or promote age acceleration?

## 2. Materials and Methods

This is a subjective, selective rather than a systematic review. PubMed abstracts were searched from the years 2008–2022 with the search term, “Accelerated Aging AND Schizophrenia” but only 1–3 research studies or reviews were selected for any one year. This was an attempt to demonstrate the evolution of concepts and research methods pertaining to the topic of age acceleration in schizophrenia. I selected papers published in successive years in high impact journals. All the cited papers are written in English, but I preferentially chose those that came from as many as possible world regions in an effort to include diverse views. 

## 3. Results

### 3.1. Is Schizophrenia a Disorder of Accelerated Aging?

Kirkpatrick and colleagues [17] suggested the possibility of accelerated aging in schizophrenia patients in 2008, citing minor physical anomalies, atypical fingerprints and palmprints, as well as metabolic and immunological abnormalities present in this disorder and ascertained prior to any drug treatment so as to avoid confounding by treatment effect. 

In 2011, Dilip Jeste’s group, experts in old age schizophrenia, wrote about accelerated physical aging in this disorder [18]. Their paper contended that, despite premature physical multimorbidity and mortality, patients with schizophrenia showed a normal rate of cognitive aging and an improvement with age of social function, as well as a decrease in psychotic symptoms. These investigators hypothesized that beneficial brain changes occur in schizophrenia over the course of aging, perhaps because of effective treatment. They acknowledged, however, that their conclusions were based on evidence that may have suffered from cohort effects (outcomes among those born in the same time period or region may not generalize to other cohorts) and survivor bias (outcomes may apply only to those who survive to old age, the most severely ill having died young). 

Two years later Chiapponi et al. [19] reviewed 26 studies of age-associated brain structure in schizophrenia. Their results speak to the heterogeneity of schizophrenia and also the heterogeneity of MRI brain scan methodology and interpretation. In summary, this review found that certain brain areas, such as the amygdala-hippocampus complex and the superior temporal gyrus appear to age normally through most of the lifespan, though they are visibly affected early in life. The uncinate fasciculus, however, which is also affected at schizophrenia onset, appears on MRI to continue deteriorating over time. 

Koutsouleris et al. [20], in 2014, compared structural brain abnormalities in schizophrenia over the aging process with those seen in other serious psychiatric disorders. The deviation from typical brain aging trajectories was defined as the difference between chronological and neuroanatomical age (as determined by a machine learning system trained on age estimation of MRI scans). Brain age was highest in the schizophrenia group, leading other diagnoses by 5.5 years. The acceleration was especially noticeable in patients who had suffered recurrent episodes of psychosis. There was no correlation in this study between brain age and illness duration, or the use of alcohol, nicotine, or any psychotropic medication.

The same year (2014), Okusaga [8] addressed the role of oxidative stress in the accelerated aging seen in schizophrenia patients. Oxidative stress, the result of free radicals released by oxygen-associated chemical reactions, directly damages both cells and connective tissue and does so indirectly as well by inducing a state of chronic inflammation. The accumulation of damaged molecules that result has been held responsible for aging progression. Patients with schizophrenia show more evidence of oxidative stress than age-matched controls although, according to the paper by Okusaga [8], too many confounders exist (antipsychotic medication, smoking, socio-economic status and unhealthy lifestyle) to attribute accelerated aging in this condition principally to oxidative stress or the associated inflammation. That same year, Shivakumar and colleagues reviewed previous studies [21] and concluded that the evidence supporting premature aging in schizophrenia was compelling. These investigators showed persuasive grouped results that demonstrated progressive changes in the brain and also in other body organs, all associated with schizophrenia. They cited studies of P300 abnormalities and visual motion discrimination, neuroimaging findings, shortened telomeres dynamics and post-mortem neuropathology. Of the possible causative factors, they hypothesized that Vitamin D deficiency, neuro-immunological changes, and high levels of oxidative stress all played contributory roles.

The part played by a shortened telomere was further examined the next year by Lin [22], Lindquvist et al. [5], and Polho [23]. Lindquvist et al. [5] concluded that the data on telomeres and schizophrenia show mixed results, and stressed the likelihood that medication exposure, and comorbidities confound the accurate interpretation of results. Lin [22] and Polho [23] suggested that effective antipsychotic treatment might be able to prevent or slow telomere shortening in schizophrenia.

In 2016, Schnak et al. [13] addressed progressive brain loss in schizophrenia. They set out to determine, via a machine learning technique, whether brain loss was a result of accelerated aging or of an unrelated process. Their conclusion was that progressive brain loss reflected two different processes: accelerated aging of the brain during the first few years of the disorder and a subsequent more variable loss that appeared to reflect individual trajectories, and were dependent, perhaps, on treatment response. 

The same year, a large multinational team [24] conducted a multisite, cross-sectional voxel-based morphometry study in first-episode and chronic schizophrenia in order to determine whether structural brain abnormalities were static (limited to the first onset time period—i.e., neurodevelopmental) or progressive (increasing over time—i.e., neurodegenerative). The design of the study enabled correlation with potential clinical moderators such as age of onset, illness duration and severity, and exposure to antipsychotics. Brain scans came from 62 first episode, 161 chronically ill and 151 control participants. First-episode schizophrenia patients showed subtle volumetric deficits relative to controls in the insula, temporo-limbic structures and striatum. Chronic schizophrenia patients showed extensive volume decreases relative to controls in bilateral superior, inferior and orbital frontal cortices, right middle frontal cortex, bilateral anterior cingulate cortices, bilateral insulae and right superior and middle temporal cortices. Correlations suggested that gray matter volume deficits were greater when onset age was early and, in some brain regions, when illness duration was long. Total gray and white matter volumes decrease correlated positively with the amount of cumulative antipsychotic exposure. The study was unable to determine the role of potential confounders such as changes in patterns of antipsychotic usage, treatment adherence, duration of untreated psychosis, substance abuse, smoking habits, cardiovascular risk factors or metabolic comorbidities.

In 2018, Dr. Jeste’s group reviewed 42 studies of potential aging biomarkers and investigated differences between persons with schizophrenia and healthy comparison volunteers [25]. They also examined the relationship of these biomarkers to age. Approximately 75% of the studies reported at least one biomarker level in schizophrenia that was significantly different from the level found in the healthy controls. Sixty-six out of 98 comparisons with controls across all marker types were positive, and the effect sizes ranged from medium to large. Differences from controls included indicators of inflammation, cytotoxicity (telomere shortening), oxidative stress, metabolic and vascular ill health, abnormal gene expression and regulation, and unusual receptor/synaptic function. Only 29% of the studies, however, showed a differential pattern of age-related decline in schizophrenia as compared to controls. Markers of receptor/synaptic function and gene expression were most frequently differentially related to age in schizophrenia while studies of telomere length and metabolic indices were least likely to indicate a faster rate of aging in schizophrenia. In fact, the studies showed a trend toward deceleration of telomeric length shortening with age in schizophrenia. With respect to metabolic risk factors, their aging effect was stronger in controls than in patients. 

Schizophrenia patients with greater disease severity and longer illness duration exhibited higher levels of inflammatory and oxidative stress biomarkers than their fellow patients, and, also, shorter telomere length. The authors of this review concluded that definitive interpretation was limited by the cross-sectional design of all the reviewed papers. They offered several potential reasons for the lack of definitive answers to the questions they posed about accelerated aging in schizophrenia: (1) cohort effects (study samples differing in illness severity and in medication use), (2) sampling effects–e.g., “healthy survivor” bias (3) nonlinear aging trajectories or presence of inflection points (pace of aging differing at different ages or different periods of illness course) (4) narrow age ranges in reported studies (5) potentially slow, gradual pace of accelerated aging in schizophrenia. They raised the possibility of current markers of aging being inadequate, and also the impossibility of separating the effects of illness duration from the effects of chronological age. They also mentioned that, given the heterogeneity of schizophrenia, some patients may well exhibit age acceleration while others may not. 

### 3.2. Current Disputes

Age acceleration in schizophrenia has become a disputed topic. In 2018, McKinney et al. [26] tested DNA methylation levels at 353 genomic sites to produce an epigenetic estimate of biological age. Using this method on three datasets (one of brain tissue and two of blood samples), they found a strong correlation with chronological age. They found no difference, however, between schizophrenia samples and those of non-psychiatric controls.

A 2020 paper in Biological Psychiatry [27] analyzed multiple epigenetic clocks that predicted chronological age, mortality, mitotic rates, and telomere length. The three mortality clocks showed acceleration in schizophrenia, driven by age-associated protein accumulation and smoking. The two mitotic clocks showed *deceleration* in schizophrenia—i.e., they predicted low cancer rates. Clozapine appeared to decelerate (by up to 7 years) multiple chronological clocks, but only in males.

In 2021, Wu et al. [28] addressed epigenetic age acceleration in schizophrenia. They employed three epigenetic clock methods to determine the epigenetic age of 1069 blood samples from patients with schizophrenia and compare them with 1264 samples from unaffected controls. They also compared 500 brain tissue samples from schizophrenia patients with 711 from controls. There were significant positive correlations between epigenetic age and chronological age in both blood and brain tissues from patients and from controls. The results showed significant epigenetic age acceleration *delay* in schizophrenia from the blood samples and from frontal cortex tissues, contradicting the accelerated aging hypothesis and supporting a neurodevelopmental model of schizophrenia as reflected by early life skewing of the epigenetic clock. 

In opposition to these findings are the results of Stone et al. in 2022 [29]. This research group found evidence of neurodegeneration correlated with increasing chronological age in schizophrenia in cognitive ability and white matter integrity. They concluded that a degenerative process was at least partially responsible for early mortality in this population.

Huang et al. [30] on the other hand, using three-modal magnetic imaging, that same year studied 138 schizophrenia patients and 208 controls aged 20–60 and found that, in schizophrenia, accelerated brain aging was only visible in the youth group. This suggests a neurodevelopmental process that affects the brain in youth and subsequently stabilizes.

In summary, up to the present day, there is a general consensus that accelerated aging, as measured by a variety of methods, is characteristic of schizophrenia, but there is substantial controversy about when this acceleration takes place. The question that appears to need answering is whether the acceleration occurs only in early life, throughout life, or only towards the end of life.

Still in 2022, Iftimovici et al. [31], using DNA methylation methods to determine epigenetic age, studied a population at ultra-high risk for schizophrenia and found that epigenetic age *decelerated* for the group as a whole, but *accelerated* in those who converted to psychosis by as much as 2.8 years when compared to non-converters. Acceleration at conversion held when age, sex, and cannabis use were controlled. Interestingly, the methylation sites affected were on genes that were linked to neurodevelopmental disorders such as schizophrenia. This study, using a completely different measure of age acceleration from that of Huang et al. [30], essentially came to an identical conclusion—that age acceleration occurs early in schizophrenia, at the time of first onset of diagnosable symptoms.

A prominent schizophrenia expert, Robin Murray, and colleagues [32], as of 2022, agree with this view—that schizophrenia is a neurodevelopmental disorder with aging genes exerting early effects on brain structure and cognitive function. According to this research team, the evidence shows that most people diagnosed with schizophrenia do not progressively deteriorate. The authors attribute the cognitive decline of some individuals with schizophrenia to the relatively poor care they receive. The Murray team position is that long term antipsychotics induce dopamine super sensitivity, making relapse and treatment resistance more likely. Long term antipsychotics can also induce cardiovascular problems, obesity, and a lifestyle that gives rise to overall poor health. The suggestion is that not everyone requires long term antipsychotics and that doses need to be kept low whenever possible and that more attention needs to be paid to the social determinants of schizophrenia [33]. Rather than the progression of the disorder itself, the prevalence of poverty, poor nutrition, and substance abuse could, according to this view, account for cognitive decline. This perspective implies that decline and premature death are not inherent aspects of schizophrenia but are, instead, secondary effects that are potentially preventable [34]. Factors that prevent deterioration may also exist in the context of schizophrenia. For instance, it has been speculated that high IQ and ample cognitive reserve protect individuals from accelerated aging [35,36]. 

A recent review from Brazil endorses the view that schizophrenia is, in principle, a neurodevelopmental disorder, but acknowledges that there is evidence, in some patients, that the process is, in addition, neuro-progressive [37] as per Kraepelin’s original concept of dementia praecox [38]. 

### 3.3. Effect of Antipsychotic Medications

A very recent paper by Du and al. [39], citing their own work on DNA methylation assays of human neuroblastoma cells cultured with various antipsychotics, conclude that antipsychotic medications *decelerate* epigenetic aging, which is what Higgins-Chen et al. [27] found with clozapine, at least in men. A large recently published consortium study of brain aging in schizophrenia-associated brain aging in 26 cohorts worldwide [40] found indices of brain age 3.6 years ahead of controls, but could not detect an effect of either usage or dosage of antipsychotics. Similarly, Talarico et al. [41] were also unable to find an antipsychotic effect on the pace of aging when studying first episode psychosis patients (FEP). They investigated telomere length in 80 FEP and DNA methylation in 60 FEP before and after a 10-week course of treatment with risperidone and compared results with those obtained from healthy controls (HC). Both indices of aging indicated *decelerated* aging compared to HC prior to treatment. Once confounders were removed, there was no further deceleration or acceleration after antipsychotic treatment, whether or not the patient’s symptoms had responded. This suggests a null effect of antipsychotics. 

## 4. Discussion

Research studies utilize several indices of progressive age (relative telomere length, volumetric shrinking of brain structures, DNA methylation) although the specific molecular pathways involved in aging are not yet understood. The process appears to be accelerated in many medical diseases, including neuropsychiatric disorders. Age acceleration is generally acknowledged in schizophrenia and has been confirmed in studies using several different indicators of aging and several different methods of ascertainment. The current debate is the timing of the process. Some studies point to accelerated aging at the time of first symptom that then normalizes after onset. The confound is that drug use, smoking, and socioeconomic disadvantage, all prevalent in schizophrenia, can produce brain changes, attributable to schizophrenia by association only. Some theorists argue that the age acceleration process, once started, continues, regardless of other factors. Some cited research only reports evidence of age acceleration towards the end of life. Precise timing remains an open question. Reckziegel et al. [37] straddle the fence. They posit that schizophrenia is a neurodevelopmental disorder that leads to early brain impairments, but that further brain deterioration may also be part of the progression of schizophrenia in some, but not all, patients. A variety of risk factors, some genetic and some environmental, may impose a neuro-progressive course. Protective factors that prevent neuro-progression may also exist. Individual trajectories differ. The role of antipsychotic medication and perhaps that of other psychotropics or anticholinergics (used to counteract parkinsonian effects of antipsychotics) remains unsettled. The limitations of this review are that it only takes into account selected representative work over successive years, starting in 2008. Many relevant studies may well have been omitted. Nonetheless, it summarizes the current state of the field and encourages further study of this important topic.

## 5. Conclusions

Schizophrenia, like many other human diseases, shows evidence of accelerated brain aging. The current thinking is that the timing of the acceleration can occur at different time periods of the lifespan in different individuals, and that antipsychotics may prevent rather than promote accelerated age. It is important to emphasize that many experts agree that cognitive decline and premature death in schizophrenia are, in principle, preventable by the application of current principles of biopsychosocial comprehensive care.

## Data Availability

Not applicable.

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
