# Peer review of "Subjective Overview of Accelerated Aging in Schizophrenia"

_ijerph, 2022, doi:10.3390/ijerph20010737_

Round 1

Reviewer 1 Report

1, Although not a systematic review, it is better to present how these literature were located;

2. Please describe how you selected the reviewed articles? By keywords or by searching the references of related articles, or...

3. Summarize the trend of the pros and against opinions of the accelerating aging for schizophrenia at the end of the argument.

  4, Please summarize the trend of selected literature to show the percentage of pro or against accelerating aging for schizophrenia

5, Add suggestion for future review or studies. What are the suggestions for future studies?

Author Response

1. Although not a systematic review, it is better to present how these literature were located

I agree with the reviewer. It would have been more thorough and authoritative to have done a systematic review. This is not what this is. I wanted to do a year by year (from 2008) review of this important field to understand better how the concept has evolved. As the years progressed, there were more and more papers addressing the question of age acceleration. I made the subjective decision to take only two or three papers in any one year. To include as many views as possible, most of the papers I chose were reviews. They were all in PubMed and all were published in high impact journals by authors with high citation indices. Because I also wanted a breadth of perspectives, I deliberately selected, when available, publications from non-Western countries. I have added the word “subjective” to the title and to the Methods section.

  1. Please describe how you selected the reviewed articles? By keywords or by searching the references of related articles, or...

I selected the papers by keying in the phrase Accelerated Aging AND Schizophrenia into PubMed but did not include all the papers that surfaced – please see above response for the papers I included.

  1. Summarize the trend of the pros and against opinions of the accelerating aging for schizophrenia at the end of the argument.

Because I am trying to show the evolution in thinking as new methods are utilized over the years since 2008, summarizing is difficult but I have tried to do that.

  4. Please summarize the trend of selected literature to show the percentage of pro or against accelerating aging for schizophrenia

Schizophrenia, like many other human diseases, shows evidence of accelerated brain aging. The current thinking is that the timing of the acceleration can occur at different time periods of the lifespan in different individuals, and that antipsychotics may prevent rather than promote accelerated age.

5. Add suggestion for future review or studies. What are the suggestions for future studies?

I would love to be able to make suggestions but I am not familiar enough with the various techniques currently used and in development used to accurately identify time periods of age acceleration and deceleration. The clinical suggestion is to apply currently known principles of biopsychosocial comprehensive care, added to the conclusion.

Thank you so much for your thorough reading and important suggestions.

Reviewer 2 Report

The author conducted a selective review (2008-2022) to investigate if schizophrenia is characterized by accelerated aging and, if so, when in the life course this might occur and if drug treatment blocks or promotes age acceleration. The paper was interesting and well written. However, failure to conduct a systematic review, with the quality of the studies identified and considered, limit the conclusions. Specifically, this is not a typical review study in that the papers were subjectively chosen, a few expert opinions were mentioned, and some of the papers were themselves review articles. No criteria were mentioned as to why the specific experts were selected, which may reflect selection bias.

The author selected the articles for the review in an attempt to show evolution and differences by area. The first aim was satisfied but the second was not, especially since the articles discussed tended to not mention where they were performed.

Abstract

Line 12. What is meant by “brief”?

Not all papers mentioned in the study had as strong of a conclusion about the relation between schizophrenia and accelerated aging as expressed by the author in the Abstract. The conclusion may be too strongly expressed.

Introduction

Lines 20-23. Missing references.

Lines 25-27. Missing commas and “(“ out of place.

Lines 34-36. Informal writing. I recommend not using the first person.

Methods

The process of selecting articles is subjective and raises the question of representation. Why were only 1-3 articles chosen per year versus more, if available? How is “high impact journals” defined?

Results

Line 173. Why is this a paradox?

Line 177. “tissue” not “tissues”

Lines 193-196. This paragraph is not referenced and seems to be out of place.

Lines 221-224. Can you make this clearer?

Discussion

Good

Conclusion

Lines 270-271. Should you add preventable through medication?

Author Response

The author conducted a selective review (2008-2022) to investigate if schizophrenia is characterized by accelerated aging and, if so, when in the life course this might occur and if drug treatment blocks or promotes age acceleration. The paper was interesting and well written. However, failure to conduct a systematic review, with the quality of the studies identified and considered, limit the conclusions. Specifically, this is not a typical review study in that the papers were subjectively chosen, a few expert opinions were mentioned, and some of the papers were themselves review articles. No criteria were mentioned as to why the specific experts were selected, which may reflect selection bias.

The reviewer is absolutely correct. It would have been more thorough and authoritative to have done a systematic review. This is not what this is. I wanted to do a year by year (from 2008) review of this important field to understand better how the concept has evolved, especially since new tools to measure acceleration and deceleration have been developed over the time period studied.  As the years progressed, there were more and more papers addressing the question of age acceleration. I made the subjective decision to take only two or three papers in any one year. To include as many views as possible, most of the papers I deliberately chose reviews. Selected papers were published in high impact journals by authors with high citation indices. Because I also wanted a breadth of perspectives, I deliberately selected, when available, publications from non-Western countries. I have added the word “subjective” to the title and to the Methods section.

The author selected the articles for the review in an attempt to show evolution and differences by area. The first aim was satisfied but the second was not, especially since the articles discussed tended to not mention where they were performed.

This is definitely a limitation. Because I screened abstracts, I could only guess at the source by the country of origin of the journal.

Abstract

Line 12. What is meant by “brief”?

I use ‘brief’ to underscore the fact that I only selected 2-3 papers in any one year

Not all papers mentioned in the study had as strong of a conclusion about the relation between schizophrenia and accelerated aging as expressed by the author in the Abstract. The conclusion may be too strongly expressed.

Thank you for pointing this out. This has been changed to “majority opinion.”

Introduction

Lines 20-23. Missing references.

Lines 25-27. Missing commas and “(“ out of place.

Thanks. Corrected.

Lines 34-36. Informal writing. I recommend not using the first person.

As pointed out by both reviewers, this is a subjective paper, which is why I use “I,” to emphasize this point.

Methods

The process of selecting articles is subjective and raises the question of representation. Why were only 1-3 articles chosen per year versus more, if available? How is “high impact journals” defined?

As mentioned, the method was, indeed, subjective. Only 203 articles were chosen because, as the years progressed, I found the number of articles overwhelming, which is why I chose so many reviews. When there were many articles from many journals, I chose those in journals that had higher impact than the others (standard high impact scores). The exception was when there were articles from relatively low income countries. I deliberately tried to choose those to address the diversity referred to earlier. As mentioned and now underscored, this was subjective.

Results

Line 173. Why is this a paradox?

I thought it was paradoxical because some of the effects of clozapine are more pronounced  in women but, admittedly, this is not the right word. It has been omitted.

Line 177. “tissue” not “tissues”

Thank you. Corrected.

Lines 193-196. This paragraph is not referenced and seems to be out of place.

This is a summary statement up to 2022. It has now be re-written to make this clearer.

Lines 221-224. Can you make this clearer?

Thank you. I have tried to do this.

Discussion

Good

Conclusion

Lines 270-271. Should you add preventable through medication?

I have changed the last line to

“..preventable by the application of current principles of biopsychosocial comprehensive care.”

Thank you so much for your valuable comments.